# A New Endemic of Concomitant Nonalcoholic Fatty Liver Disease and Chronic Hepatitis B

**DOI:** 10.3390/microorganisms8101526

**Published:** 2020-10-04

**Authors:** Hira Hanif, Muzammil M. Khan, Mukarram J. Ali, Pir A. Shah, Jinendra Satiya, Daryl T.Y. Lau, Aysha Aslam

**Affiliations:** 1Liver Center, Department of Medicine, Beth Israel Deaconess Medical Center, Harvard Medical School, Boston, MA 02215, USA; hhanif@bidmc.harvard.edu (H.H.); mkhan5@bidmc.harvard.edu (M.M.K.); mukarram4615@gmail.com (M.J.A.); pirashah2020@gmail.com (P.A.S.); jsatiya@bidmc.harvard.edu (J.S.); 2Department of Internal Medicine, University of Texas, San Antonio, TX 78229, USA; 3Department of Medicine, Louis A Weiss Memorial Hospital, Chicago, IL 60640, USA

**Keywords:** chronic hepatitis B, nonalcoholic fatty liver disease (NAFLD), nonalcoholic steatohepatitis (NASH), serum biomarkers, controlled attenuation parameter (CAP), transient elastography (Fibroscan), magnetic resonance technology, NAFLD therapy

## Abstract

Hepatitis B virus (HBV) infection remains a global public problem despite the availability of an effective vaccine. In the past decades, nonalcoholic fatty liver disease (NAFLD) has surpassed HBV as the most common cause of chronic liver disease worldwide. The prevalence of concomitant chronic hepatitis B (CHB) and NAFLD thus reaches endemic proportions in geographic regions where both conditions are common. Patients with CHB and NAFLD are at increased risk of liver disease progression to cirrhosis and hepatocellular carcinoma. Due to the complexity of the pathogenesis, accurate diagnosis of NAFLD in CHB patients can be challenging. Liver biopsy is considered the gold standard for diagnosing and determining disease severity, but it is an invasive procedure with potential complications. There is a growing body of literature on the application of novel noninvasive serum biomarkers and advanced radiological modalities to diagnose and evaluate NAFLD, but most have not been adequately validated, especially for patients with CHB. Currently, there is no approved therapy for NAFLD, although many new agents are in different phases of development. This review provides a summary of the epidemiology, clinical features, diagnosis, and management of the NAFLD and highlights the unmet needs in the areas of CHB and NAFLD coexistence.

## 1. Introduction

Non-alcoholic fatty liver disease (NAFLD) is one of the most common causes of chronic liver disease (CLD) globally and a major cause of liver-related mortality and morbidity. It consists of a spectrum of liver disease from simple steatosis, non-alcoholic steatohepatitis (NASH), progressive fibrosis, and hepatocellular carcinoma (HCC) complication [1,2]. Histologically, patients with NASH have infiltration of inflammatory cells in the hepatic lobules, hepatocyte injury with ballooning, and perisinusoidal or advanced fibrosis, in addition to macrovesicular steatosis involving ≥ 5% of the hepatocytes [2,3]. NAFLD is associated with metabolic risk factors such as obesity, dyslipidemia, and type 2 diabetes mellitus [2]. Owing to the trend of sedentary lifestyle and globalization of the Western diet, the prevalence of NAFLD has increased exponentially at about 114% for men and 80% for women between 2004–2016 [4,5]. Hepatitis B virus (HBV) infection is another CLD of global magnitude; it is a significant cause of cirrhosis, hepatocellular carcinoma, grave morbidity, and mortality. Since both chronic hepatitis B (CHB) and NAFLD are common liver conditions that can lead to end-stage liver disease (ESLD) and HCC, it is important to understand the impact of NAFLD on hepatitis B and vice versa. Recent studies reported that concomitant presence of HBV and hepatic steatosis is associated with increased risk of disease progression to cirrhosis and hepatic and extra-hepatic malignancies [6,7,8]. In this review, we discuss the epidemiology; natural history; diagnostics, including non-invasive biomarkers and radiological modalities; and management strategies for NAFLD among patients with chronic hepatitis B.

## 2. Epidemiology

Globally, the cumulative prevalence of NAFLD is estimated to be more than 1 billion of the world population [9,10]. There has been a notable rise in NAFLD disease burden in the past decade, with the highest rates in the Middle East (32%), followed by South America (31%), Asia (27%), the USA (24%), and Europe (20-24%) [9,10,11] (Figure 1, top). In the USA, the prevalence of NAFLD increased from 15% in 2005 to approximately 25% in 2017 [12]. Similarly, an analysis of 237 studies from Asia noted that the prevalence of NAFLD increased from 25.3% in 2005 to almost 34% in 2017 [13]. The increased prevalence of NAFLD has been attributed to the surging rates of obesity, insulin resistance, and socioeconomic and lifestyle factors. Over 95% of obese individuals undergoing bariatric surgery and > 70% of patients with type 2 diabetes mellitus have NAFLD [14]. The presence of NAFLD varies in different racial and ethnic populations, with the highest prevalence among Hispanics (45% to 58%), followed by non-Hispanic Whites (33%) and African Americans (24%) [15]. NAFLD has become the second most common cause of liver transplantation in the USA and the most common liver disorder in Western countries [4,5].

It is estimated that 240 to 350 million of the world population has hepatitis B [16] (Figure 1, bottom). NAFLD is common in some of the HBV-endemic regions such as Asia. North America is considered to have low HBV prevalence, with an estimated rate of 2.2 million in the United States. This is likely an underestimation due to the influx of immigrants from high HBV-endemic regions [17]. Since NAFLD and hepatitis B are common, it is frequent for an individual to have both liver diseases [18]. According to a meta-analysis by Hui et al. including 17 studies, the prevalence of hepatic steatosis was about 25 to 30% among patients with CHB [19]. 

## 3. Clinical Manifestations in Patients with Concurrent NAFLD and CHB

With an alarming increase in the number of patients with concomitant NAFLD and CHB [20,21,22,23,24], there is an urgent need to evaluate the interactions between these two diseases and the roles of hepatic steatosis (HS) and NAFLD on the prognosis of hepatitis B. There are also some controversial reports on the steatogenic effects of HBV in in vitro models. Wu and colleagues reported that HBV X protein (HBx) might promote hepatic lipid accumulation through up-regulating fatty acid binding protein (FABP1) [25]. Another study by Kim et al. determined that hepatic lipogenesis was mediated by HBx up-regulation of sterol regulatory element-binding protein 1 (SREBP-1) and peroxisome proliferator-activated receptor gamma (PPAR-γ) [26]. Conversely, clinical studies showed that host metabolic factors rather than viral factors are important in the development of HS in CHB [27,28]. In the aforementioned studies, high body mass index (BMI), hypertriglyceridemia, increased waist circumference, and insulin resistance were found to be more prevalent in CHB patients with steatosis compared to those without steatosis.

In a retrospective study conducted in a military hospital in Turkey, Karacaer et al. investigated the influence of HS on necroinflammation and fibrosis in 254 young male CHB patients with and without HS. They found that patients with HS had significantly more fibrosis (*p* = 0.012; *r* = 0.158) and higher histologic activity index (HAI) (*p* = 0.029; *r* = 0.137) [29]. Similar results were reported by Estakhri and colleagues: abnormal serum aminotransferases, increased fibrosis, and inflammation were noted in CHB patients with concurrent NAFLD [30]. It is important to note that both the aforementioned studies showed no relationship between HBV DNA titers and severity of liver disease among those CHB patients with NAFLD [29,30]. Charatcharoenwitthaya et al. identified steatohepatitis as an independent predictor of significant fibrosis (METAVIR score, F ≥ 2) odds ratio (OR) 10.0; 95% confidence interval (CI, 2.08–48.5) and advanced fibrosis (METAVIR score, F ≥ 3) (OR, 3.45; 95% CI, 1.11–10.7) after adjusting for HBV DNA levels and metabolic syndrome [31]. On the contrary, some studies did not observe a correlation between HS and disease severity in CHB patients [32,33,34,35]. These discrepancies can be partially explained by the small sample size, heterogeneous patient populations, and retrospective nature of the study design.

In a large retrospective multiethnic North American and European cohort study, Choi et al. reported that patients with CHB and NASH had higher liver-related outcomes and overall mortality than those with CHB alone [36]. They proposed a “two-hit theory” for the pathogenesis of disease progression for those with both NASH and CHB. In this theory, HBV is considered a first hit in inducing hepatocyte injury, the superimposed NASH serves as the second hit leading to progressive hepatic inflammation and fibrosis. This theory is substantiated by the observation where the lipid-laden hepatocytes induce mitochondrial and peroxisomal dysfunction that result in oxidative stress, toxic fatty acid metabolite production, and hepatocyte damage [37] The ongoing chronic inflammation with activated myofibroblasts derived from hepatic stellate cells can lead to increased extracellular matrix (ECM) deposition [37,38]. Hence, CHB and NASH can synergistically cause liver disease progression.

CHB is the most important risk factor for HCC, accounting for about 50% of the HCC cases globally. The HCC risk in HBV endemic areas is even higher and contributes to 70–80% of the cases [39]. Patients with NASH-related cirrhosis are also at a greater risk for HCC with a yearly cumulative incidence of 2.6% [40]. In a retrospective cohort study, Chan and colleagues in Hong Kong evaluated the HCC risk in CHB patients with concurrent NAFLD [24]. Out of 270 CHB patients, 107 had concomitant hepatic steatosis. Among them, nine patients with fatty liver (8.4%) and two patients without fatty liver (1.2%) developed HCC, with a hazard ratio of 6.84 (95% CI: 1.48–3.66; *p* = 0.014). Similar results were found in a study conducted by Lee et al. in a total of 321 CHB patients, wherein a threefold increased risk of HCC was observed among those with NAFLD [22]. It is increasingly recognized that metabolic factors, the precursors of NAFLD, also contribute to HCC development in CHB [41,42]. Yu et al. evaluated 1690 Taiwanese patients with CHB and a positive correlation was established between metabolic factors and HCC; there was a significantly increased risk of HCC for those with three or more metabolic risk factors compared to none (HR, 5.06; 95% CI, 2.23–11.47) [43]. Insulin resistance and obesity were recognized as two of the most significant risk factors for HCC in CHB [28,44,45].

## 4. Current and Novel Serum Biomarkers for NAFLD

The interplay of NAFLD and CHB is complex; both disease entities can cause elevation of serum aminotransferases. This may pose diagnostic and management challenges in patients with combined disease. Demir et al. observed that NAFLD rather than CHB may be the most common cause of elevated alanine transaminase (ALT) levels among patients with HBeAg-negative CHB and HBV DNA levels < 2000 IU/mL [46]. It is desirable to have biomarkers that can identify NAFLD among CHB patients for timely management of both liver conditions [8]. Liver biopsy is the gold standard to diagnose NAFLD and NASH, however, there are several limitations including its invasive nature, cost, sampling error, and inter-observer variability among pathologists [47].

### 4.1. Serum Biomarkers for Diagnosing Hepatic Steatosis, NASH, and Fibrosis

#### 4.1.1. Serum Biomarkers to Detect Hepatic Steatosis

There are a number of non-invasive biomarker assays available to detect hepatic steatosis. They include the Fatty Liver Index (FLI) [48], Hepatic Steatosis Index (HSI) [49], NAFLD Liver Fat Score [50], Steato Test [51], and NAFLD Ridge Score [52] (Table 1). The HSI, NAFLD Liver Fat Score, and NAFLD Ridge Score have been specifically developed to detect steatosis due to NAFLD, whereas the FLI and Steato Test can be applied to detect steatosis due to any cause. FLI and HSI are feasible in different clinical settings as both are derived from common laboratory parameters. They have good accuracy when measured against ultrasound [48,49]. They have not been evaluated using liver biopsy. The NAFLD Liver Fat Score provides a quantitative value for steatosis and has high accuracy with area under receiver operating characteristic curve (AUROC) of 0.86–0.87; it may not be feasible for routine clinical use as serum insulin is required in its algorithm [50]. The Steato Test and NAFLD Ridge Score combine multiple parameters to predict hepatic steatosis and have been used in many cross-sectional studies [51,52]. Their ability to detect changes in liver fat content over time is limited. The Steato Test has modest accuracy, with AUROC of 0.79–0.80, but has good prognostic value in patients with diabetes mellitus (DM) and dyslipidemia. The presence of severe steatosis by Steato Test is associated with a twofold increase in all-cause mortality and cardiovascular mortality. However, it may not be feasible in some regions due to its high cost. The NAFLD Ridge Score has very good accuracy, with AUROC of 0.87, and has an excellent negative predictive value of 96% to exclude NAFLD; however, its use is currently limited to research settings.

#### 4.1.2. Serum Biomarkers to Detect NASH

NASH is associated with worse liver disease prognosis comparted to simple hepatic steatosis. It is therefore important to identify patients with NASH to intensify interventions and to prevent further disease progression to advanced fibrosis. The liver injury in NASH is complex and involves pathological processes such as hepatocellular apoptosis, inflammation, oxidative stress, and abnormal adipokine signaling [53]. Various noninvasive serum biomarkers for NASH based on different pathophysiologic mechanisms have been developed. They include apoptosis markers cytokeratin-18 (CK18) fragments [54], total cytokeratin [55] and soluble cell surface receptor (sFAS) [56], inflammatory biomarkers such as C-reactive protein (CRP) tumor necrosis factor (TNF) interleukin-8 (IL-8) C-X-C motif chemokine ligand 10 (CXCL10) [56], lipid peroxidation products (11-hydroxyeicosatetraenoic acids {11 HETE} 9-hydroxyoctadecadienoic acid {9-HODE} 13- hydroxyoctadecadienoic acid {13-HODE} oxo-octadecadienoic acid {13-oxo-ODE} linoleic acid 13-hydroxyoctadecadienoic acid {LA-13-HODE}) [57,58], adipocytokines and hormones (adiponectin, leptin, resistin, visfatin, retinol binding protein 4 {RBP4}, fatty acid binding protein-4 {FABP4}, fibroblast growth factor 21 {FGF21}) [59,60], lysosomal enzymes (cathepsin D) [61], and combined tests such as the NASH Test [62] and NASH Diagnostic Panel [63] (Table 2). CK18 is commercially available and was noted as correlating with histological improvement in NASH. When combined with sFAS, the diagnostic accuracy of CK18 is higher but the optimal predictive cut-off value remains uncertain. Various inflammatory biomarkers have been used to correlate the inflammatory activity in NASH. The inflammatory markers are not suitable to diagnose NASH specifically as their levels are influenced by other systemic inflammatory conditions [56]. The measurement of lipid peroxidation products requires mass spectroscopy and has not been extensively validated for NASH [57,58]. Combination of adipocytokines has been applied to validate NASH in the bariatric population but requires further research and validation in other patient cohorts. Their accuracy may be influenced by visceral adiposity besides hepatic steatosis [59,60]. Due to limitations of individual markers for NASH, combined panels such as the NASH Test and NASH Diagnostic Panel were formulated as proprietary algorithms to increase the predictive accuracy. NASHTest comprises of age, gender, height, weight and serum levels of triglyceride, cholesterol, α2m, apolipoprotein AI, haptoglobin, Gamma-glutamyl transferase (GGT), alanine transaminase (ALT), aspartate transaminase (AST), and total bilirubin [62,63]. The NASH Diagnostic Panel includes diabetes mellitus status, gender, BMI, serum levels of triglycerides, CK18 fragments, and total CK18. The AUROC values for the NASHTest and NASH Diagnostic Panel were 0.69–0.79 and 0.81, respectively, in predicting NASH. Both combined panels have been tested in relatively small number of patients and in predominantly bariatric populations.

#### 4.1.3. Serum Biomarkers to Detect Fibrosis in NAFLD

Progression of NASH to fibrosis, especially > F2 fibrosis, is associated with increased morbidity and mortality [64]. Several non-proprietary biomarkers and panels have been evaluated for the detection of fibrosis. These include the AST/ALT ratio [65], AST/Platelet Ratio Index (APRI) [66], Fibrosis 4 (Fib 4) Index [67], NAFLD Fibrosis Score [68], and BARD Score [69] (Table 3). The AST/ALT Ratio Index and APRI have low accuracy in diagnosing fibrosis. Fib 4 Index has modest accuracy, with AUROC of 0.83 for detecting F3 fibrosis, which is comparable to the NAFLD Fibrosis Score [50,66,67]. The NAFLD Fibrosis Score, which comprises age, BMI, impaired fasting glucose, AST, ALT, platelet count, and albumin, has high accuracy, with AUROC of 0.83. It has been extensively validated and has been used as a predictor of liver decompensation and mortality in patients with NAFLD [69]. A major limitation in using the NAFLD Fibrosis Score and BARD Score is the varied interpretations of BMI across the different ethnic groups [68,69]. Several specific fibrosis markers and panels are also available. These include hyaluronic acid (HA) [70], N-terminal propeptide of procollagen type III (PIIINP) [71], Pro-C3 [72], tissue metalloproteinases 1 (TIMP1) [73,74], laminin [75], enhanced liver fibrosis (ELF) [76], FibroTest [77], and FibroMeter NAFLD [78,79]. Most of the biomarkers have a high negative predictive value of > 90% in ruling out fibrosis. The challenge is the inability of these makers to differentiate between different stages of fibrosis. 

There are currently no official guidelines to diagnose and manage NAFLD in patients with CHB. Most of the biomarkers applied to detect steatosis, inflammation, and fibrosis for NAFLD, as mentioned above, have not been validated for patients with underlying CHB. Moreover, many of those biomarker panels for steatosis and fibrosis include ALT and AST. The fact that ALT and AST are frequently elevated in both NAFLD and CHB means that these biomarkers may not be applicable for patients with both liver diseases. Novel noninvasive serological biomarkers and panels that have been evaluated in patients with concurrent CHB and NAFLD are summarized below.

### 4.2. Serum Biomarkers for Diagnosing Hepatic Steatosis, NASH, and Fibrosis in CHB Patients

#### 4.2.1. Serum Biomarkers to Detect Hepatic Steatosis in CHB Patients

Routine clinical parameters to assess risks of NAFLD in CHB patients include fasting glucose, total cholesterol (TC), and triglycerides (TGs). A Hepatic Steatosis Model consisting of BMI, hemoglobin (Hb), TGs, and serum uric acid was applied to CHB patients in China [77]. It had good accuracy, with AUROC of 0.84 in comparison to a threshold of 22% of steatosis on liver biopsy. Large multicenter cohort studies are required to further validate the results. Ou et al. applied a Fatty Liver Test composed of diastolic blood pressure, weight, and waistline on 1312 Asian CHB patients [78]; 618 (47%) of them had steatosis by controlled attenuation parameter (CAP). The AUROC for detecting steatosis was 0.79 in the training phase, which was comparable to 0.82 in the validation phase. The advantage of the Fatty Liver Test is its feasibility. CAP, however, is still considered a research tool. Before it can be recommended for clinical use, it would require validation, preferably using liver biopsy and histologic quantification in larger patient cohorts.

There are a number of gene polymorphisms, especially patatin like phospholipase domain containing 3 (PNPLA3) which have been shown to have genetic predisposition to NAFLD. PNPLA3 gene codes for adiponutrin, a protein located in both adipocytes and liver cells. Pan et al. evaluated PNPLA3 polymorphisms in Chinese CHB patients with and without biopsy-proven NAFLD. Four single nucleotide polymorphisms (SNPs) of PNPLA3 (rs738409 G, rs3747206 T, rs4823173 A, and rs2072906 G alleles) were found to be linked to a high risk of NAFLD in CHB after adjusting for age, sex, and BMI. These genotypes were also associated with progression from steatosis to NASH and liver fibrosis. These important observations need to be validated in patients with different racial and ethnic backgrounds. PNPLA3 polymorphism was found to downregulate HCV replication in human studies and mouse models [80]. The effects of these PNPLA3 SNPS on HBV pathogenesis are not well understood.

#### 4.2.2. Serum Biomarkers to Detect NASH in Patients with CHB

A number of noninvasive score models were applied to identify NASH among patients with CHB. In a relatively small study with 64 CHB patients, Liang et al. found that elevated serum level of CK18M30, higher CAP scores, and elevated fasting plasma glucose level were all independent risk factors for NASH [81]. Subsequently, a logistic regression model combining CK-18 M30, CAP, fasting plasma glucose, and HBV DNA level was established for diagnosis of NASH in CHB patients. Multicenter studies with larger cohorts are required to validate this score model for clinical utility [81].

Another active research area to identify NASH in CHB patients is lipidomic profile assessment. Yang et al. applied ultra-performance liquid chromatography–tandem mass spectrometry to determine lipidomic profiles between CHB patients with and without NASH [82]. Most of the serum ceramide and neutral lipids were elevated in the NAFLD group. Serum monounsaturated triacylglycerols (TAG) were significantly increased in NASH subjects (OR = 3.215; 95% CI 1.663–6.331) and correlated positively with histological activity (*r* = 0.501, *p* < 0.001). A limitation of the study is the lack of simultaneous evaluation of liver tissue. It is important to validate these results in CHB patients with mild to severe liver fibrosis.

Micro RNAs (miRNAs) are small noncoding protein transcripts that may be involved in many physiological and cellular mechanisms. Circulating levels of these miRNAs are stable, sensitive, and specific to serve predictors of various pathological processes. Zhang et al. applied several miRNAs to differentiate liver injury caused by hepatitis B and NASH compared to healthy control subjects [83]. There were 34 differential expressions of miRNAs in patients with CHB compared to control subjects. They identified higher levels of miRNA: -122, -638, -572, and -575, and lower levels of miRNA: -744, in the sera of patients with CHB and NASH, respectively. In another study, Liu et al. assessed a panel of miRNAs for NASH diagnosis [83]. They found that MiRNA-34a was a more accurate marker for the diagnosis of NASH compared to CK-18, FIB-4 (Fibrosis index based on **4** factors), and APRI, with AUROC of 0.811 (95% CI: 0.670–0.953). These studies demonstrated that miRNAs may serve as potential biomarkers for NAFLD and HBV-associated liver injury, but they need to be carefully validated in different patient populations [83,84].

#### 4.2.3. Serum Biomarkers to Detect Fibrosis in NAFLD and CHB Patients

As none of the commonly available fibrosis biomarkers have been validated for patients with concurrent CHB and NAFLD, Dong et al. applied the Forns Index, consisting of platelet count, gamma glutamyl transpeptidase (GGT), age and cholesterol levels, Fibroscan, and acoustic radiation force impulse (ARFI), alone and in combination, to evaluate hepatic fibrosis in CHB patients. In addition, the authors assessed the effects of inflammation and steatosis on the accuracy of these diagnostic methods. A total of 81 patients were included in the study, with liver biopsy as the standard for comparison. They concluded that the combination of Foms index with Fibroscan or ARFI increases the accuracy of diagnosing fibrosis compared to the individual modality. Inflammation, but not steatosis, may affect the diagnostic accuracy of these methods [85]. Lemoine and colleagues reported that a novel fibrosis model utilizing gamma glutamyl transpeptidase-to-platelet ratio (GPR) may be better than APRI (aspartate aminotransferase-to-platelet ratio index) in diagnosing cirrhosis [86]. This result was not substantiated by a latter study including patients with CHB and NAFLD [87]. Ozlem et al. evaluated the role of soluble urokinase plasminogen activator (uPAR) in determining severity of liver fibrosis in patients with chronic hepatitis B or C and NAFLD. Plasma levels of uPAR were significantly elevated in those with hepatitis B or C compared to healthy controls, but the uPAR levels in patients with NAFLD were similar to healthy controls. [88] It is unknown how this marker will perform among patients with both chronic viral hepatitis and NAFLD.

## 5. Imaging Modalities for Detection of NAFLD in CHB

### 5.1. Imaging Modalities for Detection of Hepatic Seatosis

Due to rising prevalence of NAFLD, it is important to develop and evaluate non-invasive methods for the diagnosis of the wide spectrum of NAFLD from steatosis to NASH and fibrosis. Liver biopsy is considered a gold standard diagnostic tool, but it is an invasive procedure with potential complications such as pain and bleeding. Imaging modalities have gained a better patient acceptance and have been used as a substitute for the detection of steatosis and fibrosis [89].

#### 5.1.1. Abdominal Ultrasound

Ultrasound as been applied widely as a screening tool for hepatic steatosis due to its accessibility. In a study conducted by Mottin et al., the sensitivity and specificity of ultrasound in detecting hepatic steatosis were only 49% and 75%, respectively, using liver biopsy as a reference in a cohort of morbidly obese patients [90]. Besides the presence of obesity, ultrasound has reduced accuracy for detection of hepatic steatosis in patients with significant fibrosis or renal disease [91].

#### 5.1.2. Controlled Attenuation Parameter (CAP)

Controlled attenuation parameter (CAP) is a recently developed software that measures ultrasound attenuation. It is implemented on Fibroscan, an ultrasound-based vibration controlled transient elastography device that is used to assess the elasticity of the liver. An earlier study provided encouraging results, showcasing that CAP could detect > 11%, > 33%, and > 66% hepatic steatosis with AUROCs of 0.91, 0.95, and 0.89, respectively. There were, however, significant overlapping CAP scores for the different degree of steatosis using the standard M-probe, especially for obese individuals [92,93]. The XL probe was later developed for obese patients. The resulting CAP scores with the individualized M probe and XL probe had similar performance in terms of AUROC and cut-off values. One must practice caution when interpreting CAP scores relative to the timing of food, as CAP scores increase after meals across all stages of fibrosis [94].

Since our review focuses on NAFLD and CHB, it is quite relevant to compare the performance of CAP to ultrasound in patients with CHB. Liang et al. evaluated 366 CHB patients and found CAP had better diagnostic accuracy compared to conventional ultrasound. The rate of steatosis overestimation, however, was significantly greater for CAP than for ultrasound (30.5% vs. 12.4%) [95]. CAP is an attractive and efficient point-of-care assessment tool but needs to be further validated, especially in patients with concomitant chronic liver disease such as CHB.

#### 5.1.3. Magnetic Resonance Technology

Magnetic resonance (MR)-based techniques such as magnetic resonance elastography (MRE) and proton density fat fraction (PDFF) have been shown to accurately diagnose steatosis and fibrosis. MRE utilizes propagating mechanical shear waves to assess the stiffness of the tissue. Waves propagate faster in stiffer tissues as compared to softer tissues. The speed of propagation is reflected on wavelength as increased stiffness of tissue associated with longer wavelengths. Imaging of these waves are performed with a magnetic resonance imaging (MRI) sequence and processed to generate elastograms [96]. Proton density fat fraction (PDFF) is another non-invasive tool that utilizes triglyceride-specific signal intensity and offers a sensitive approach for steatosis detection. It has been developed to specifically assess fat over the entire liver [97]. It determines the ratio of the density of mobile protons from the triglycerides and the total density of protons from mobile triglycerides and mobile water. It has been validated in multiple studies with different racial cohorts of NAFLD patients. In those studies, MRI-PDFF outperformed CAP in terms of AUROC and accuracy across all grades of steatosis [98,99,100]. In a cross-sectional study of 100 patients, Park et al. systematically showed that MRI-PDFF was more accurate than CAP in diagnosing all grades of steatosis with liver biopsy as the reference standard [99] (Table 4). MRI-PDFF, however, is not suitable for routine clinical settings due to its high cost and requirement of specific facility infrastructure.

### 5.2. Imaging Modalities for Detection of NASH

Different imaging modalities have been tried and tested for the diagnosis of steatohepatitis. Studies using transient elastography such as Fibroscan and magnetic resonance technology to assess NASH have shown a wide range of AUROCS, with the optimal cut-off being affected by the degree of fibrosis. In a study by Imajo et al., magnetic resonance imaging had a higher capability of diagnosing NASH as compared to transient elastography (TE), with AUROC 0.91 vs. 0.82 [100]. Recently, Naganawa et al. studied the role of non-contrast enhanced CT in the diagnosis of NASH and found that in patients without fibrosis, it has high sensitivity (100%) and specificity (92%) in diagnosing NASH, with an AUROC value of 0.93-0.94. For patients with high hyaluronic acid and hepatic fibrosis, however, the sensitivity (42%) and specificity (31%) was significantly reduced [101]. Eddowes et al. evaluated the role of multiparametric MRI (liver multiscan) to differentiate simple steatosis (SS) from NASH. The results were suboptimal with AUROC for SS at 0.69 (0.50–0.88) and AUROC for NASH at 0.74 (0.59–0.89) [102]. All these studies have shown promising results but have limitations, especially in the presence of variable amount of hepatic fibrosis. Liver biopsy remains the gold standard for diagnosing NASH [103].

### 5.3. Imaging Modalities to Assess Fibrosis

Two types of imaging techniques have been applied. First is ultrasound-based technology, which measures the speed of the shear waves provoked in the liver tissue. On the basis of the generation and detection of these shear waves, researchers have invented different ultrasound-based elastography techniques, including transient elastography, supersonic shear wave elastography, and acoustic radiation force impulse (ARFI) elastography. The second imaging technique is magnetic resonance-based imaging, which utilizes a magnetic resonance scanner to detect the difference in MR frequency between the protons in fat and water.

#### 5.3.1. Ultrasound-Based Elastography Modalities

Transient elastography (TE), measured by Fibroscan, has emerged as one of the most commonly used technologies to access hepatic fibrosis. Several meta-analyses have shown that it has an accuracy of 88–89% and 93–94% for diagnosing advanced fibrosis and cirrhosis, respectively [104,105,106,107,108]. The vibration-controlled transient elastography (VCTE) used in the Fibroscan device provides readily available results with convenient examination and clinical settings. Currently, transient elastography is recommended for the assessment of fibrosis in NAFLD [109]. Two-dimensional shear wave elastography (2D-SWE) is another ultrasound-based technology with promising results. In one meta-analysis, it showed a diagnostic accuracy of 93% and 92% for advanced fibrosis and cirrhosis, respectively, in NAFLD patients [110]. ARFI measures a beam that passes over a standardized region of the liver [111]. ARFI had poor diagnostic performance in patients with a BMI greater than 35 kg/m^2^ [112]. Both 2D-SWE and ARFI techniques need further prospective studies to validate their performance in NAFLD patients and are not recommended currently in management guidelines.

#### 5.3.2. Magnetic Resonance-Based Elastography Modalities

A meta-analysis by Singh et al. reported that magnetic resonance elastography (MRE) has a very high accuracy for detecting advanced fibrosis and cirrhosis, regardless of BMI and etiology [113]. This makes MRE a very appropriate modality if we want to access disease progression and treatment response in patients with chronic liver diseases. In a cross-sectional study of 100 patients, MRE was more accurate in assessing the degree of fibrosis in patients with NAFLD as compared to TE (Fibroscan). Receiver operating characteristic curves (ROC) were used to assess the performance of MRE compared to Fibroscan in diagnosis of fibrosis with liver biopsy as a reference. MRE was found to be more accurate in detecting any fibrosis versus no fibrosis compared to Fibroscan (*p* = 0.01). There was no significant difference between MRE and Fibroscan for accurate diagnosis in between various stages of fibrosis [99] (Table 5).

## 6. Management of Patients with HBV and NAFLD

To date, there is no U.S. Food and Drug Administration (FDA)-approved treatment available for NAFLD. The current management options aim at lifestyle modifications with a goal of weight reduction. Medications, including anti-diabetics, are being investigated and repurposed for NAFLD. There are many novel pharmacological agents in clinical trials, with the goal being to reduce inflammation and fibrosis in NASH. The American Association for the Study of Liver Disease (AASLD) currently recommends pharmacological treatment only for the patients with biopsy-confirmed NASH and fibrosis [2].

### 6.1. Lifestyle Modifications for NAFLD

Although there have been many breakthroughs in understanding epidemiology and pathophysiology of NAFLD, weight loss remains the cornerstone treatment for NAFLD [94]. The AASLD suggests a weight loss goal of 3–5% of total body weight for improvement in steatosis. In addition, 7–10% weight reduction is required to improve fibrosis and other histological features of NASH [2,114]. Weight loss can lead to remission of NAFLD in patients with a BMI < 25%. [115].

Generally, low calorie diets are recommended for patients with NAFLD/NASH [94]. Nguyen, V. and George, J. recommend a hypo-caloric diet (1200–1500 kcal/day in normal weight or 500–1000 kcal/day in overweight populations), with the aim of achieving 5–7% reduction in baseline weight over a year for NAFLD management; this resulted in histological improvement of steatosis and steatohepatitis [116,117].

Intake of simple carbohydrates is associated with NASH development. Fructose, being a simple carbohydrate, is associated with increased hepatic fibrosis due to its rapid metabolism in the liver, leading to a decrease in hepatic ATP level and hepatic oxidative damage [118,119]. Polyunsaturated fatty acids (PUFAs) such as n-3 PUFA have been reported to reduce systemic inflammation and oxidation [120,121]. The WELCOME study demonstrated a mild reduction in liver fat, with omega-3 fatty acids used at a dose of 4 g/day for 15 to 18 months, but did not result in an improvement in fibrosis scores [122].

It is important to note that macronutrient-specific diets have not been as efficacious as total caloric restriction for NAFLD. Many dietary regimens such as the Mediterranean diet and Dietary Approach to Stop Hypertension (DASH) have been proposed to control NAFLD. A case control study carried out by Hekmatdoost et al. showed that adherence to a DASH diet is negatively associated with risk of NAFLD development [123]. The European Association for the study of the Liver (EASL), European Association for the Study of Diabetes (EASD), and European Association for the Study of Obesity (EASO) recommends the Mediterranean diet for the treatment of NAFLD [109]. The Mediterranean diet consists of whole grain cereals with a low or medium glycemic index (GI ≤ 55), polyunsaturated and monounsaturated fatty acids (PUFA and MUFA, respectively) from olive oil, phytochemicals and antioxidants, and a moderate amount of wine, having been shown to have a positive impact in NAFLD through early satiety and reduction in hepatic steatosis and inflammation [124,125]. Houghton et al. reported that exercise, independent of weight loss, confers beneficial effects in patients with biopsy-proven NASH. Both aerobic and resistance exercises have been implicated to have equal benefits by decreasing intra-hepatic triglyceride and increasing insulin sensitivity [126].

The amount of weight loss required to improve histological features of NASH is difficult to achieve and is maintained by lifestyle interventions [127]. A weight loss of more than 1.6 kilograms (kgs) per week can cause a paradoxical increase in the state of inflammation within the liver [128]. An additive benefit has been reported when cognitive behavioral therapy was used in conjunction with weight loss [129].

For selective patients, bariatric surgery leads to improve co-morbidities, decreased cardiovascular-related deaths, and all-cause mortality in addition to weight loss [130]. In a prospective study, laparoscopic sleeve gastrectomy was associated with improvement of NASH, hepatocyte ballooning, steatosis, lobular inflammation, and even fibrosis at 1 year after surgery [131]. Similar benefits such as weight reduction, improved ALT, GGT, steatosis, hepatocyte ballooning, lobular inflammation, and fibrosis at 1 year post-bariatric surgery were observed in another study [132]. The AASLD does not recommend bariatric surgery for the treatment of NASH alone. It may be considered in eligible patients with morbid obesity and NASH [2]. More prospective randomized controlled trials are required to further evaluate the long-term effects of bariatric surgeries. Coffee consumption can slow disease progression and has been showed to have a protective effect in patients with NAFLD [133,134]. Alcohol consumption should be avoided in these patients, especially those with concurrent obesity, as it increases the risk of hepatocellular carcinoma (HCC) [135].

### 6.2. Therapeutic Agents in Development for NASH

Pharmacological management of NAFLD is recommended for subgroups of patients with progressive NASH, early-stage NASH with high risk of fibrosis (age >50 years, metabolic syndrome, diabetes, increased ALT), or active NASH with necroinflammatory activity [2]. The AASLD, EASL, EASD, and EASO recommended the use of pioglitazone and vitamin E for the treatment of biopsy-proven NASH in patients with or without type II diabetes mellitus, keeping in consideration the side effects of pioglitazone and vitamin E [2,109]. A randomized control trial comparing the efficacy of vitamin E, pioglitazone, and placebo for patients with type 2 diabetes mellitus and NASH did not show any improvement in terms of vitamin E alone or placebo. Combination therapy with vitamin E and pioglitazone reduced the NAS score by > 2 points and resolution of NASH in 33% of patients [136,137]. Treatment with vitamin E carries minimal but serious side effects, including hemorrhagic stroke and prostate cancer [138,139].

Current recommendations from the AASLD and EASL recommend vitamin E to be used at a dose of 800 IU daily as a short-term option for nondiabetic adults with biopsy-proven NASH [2,140].

The therapeutic use of ursodeoxycholic acid (UDCA) in NAFLD/NASH has been ascribed to its cytoprotective effects through modulation of mitochondrial pathways to reduce apoptosis [141]. There are conflicting results in terms of its beneficial outcomes, despite early promise. Pentoxifylline, a TNF-α inhibitor, increases hepatic glutathione synthesis and decreases the production of free oxygen radicals. Owing to its antioxidant effects, it has shown to be hepatoprotective, resulting in improvement in liver histology [142].

Antidiabetic drugs such as glucagon-like peptide 1 (GLP-1) agonists have been found to be effective in NASH. In a randomized, double-blinded, placebo-controlled LEAN trial (Liraglutide Efficacy and Action in NASH), 1.8 mg liraglutide for 48 weeks achieved resolution of NASH in 39% of patients compared to 9% in the placebo-treated group (*p* = 0.019). Fibrosis progression was noted in only 9% of the liraglutide-treated patients compared to 36% in the placebo group (*p* = 0.04) [143]. In a prospective non-randomized, non-placebo-controlled 6-month therapy with 1.2 mg liraglutide in patients with uncontrolled type II diabetes improved hepatic fat content, as measured by hepatic magnetic resonance spectroscopy (^1^H-MRS). The reduction in steatosis was highly correlated with weight reduction in these patients (*r* = 0.490, *p* < 0.0001). In contrast, patients with similar baseline characteristics who were managed with insulin therapy had no change in BMI and liver fat content [144]. Despite the promising results from recent trials, more randomized controlled trials are needed for liraglutide to be recommended for NASH. Several trials have evaluated the use of metformin in patients with biopsy-proven NAFLD. Although it initially decreased liver enzymes and improved insulin sensitivity, it did not result in histological improvement [145,146,147,148].

Sitagliptin, a dipeptidyl peptidase 4 (DPP-4) inhibitor, had conflicting results regarding improvement in NASH. In an open, randomized controlled clinical trial, 46 patients with NASH were advised to have lifestyle modifications and 23 patients were given sitagliptin 100 mg once daily for 1 year. Paired biopsies of 40 patients who completed the trial showed that sitagliptin was associated with improvement in steatosis and NAS compared to the control group. [149]. A prospective study with a larger patient sample size is required to confirm the benefits.

Sodium glucose transport protein 2 (SGLT2) inhibitors are used in the treatment of type II diabetes mellitus and were evaluated in NAFLD [150,151]. Empagliflozin, at a dose of 25 mg/day for 24 weeks, was used in an open-label pilot study involving nine diabetic patients with biopsy-proven NASH. Histological features were unchanged or improved in all but one patient who had evidence of worsening of hepatocellular ballooning [152]. Large, randomized placebo-controlled trials including histological endpoints by biopsies are warranted to learn the definitive benefits of SGLT2 inhibitors in patients with NASH with or without diabetes.

Farnesoid X receptor (FXR), a receptor found in the nucleus of liver cells and the intestine, is a key regulator of bile acid metabolic pathways [153]. FXR increases the expression of small heterodimer protein (SHP) 1, which is inhibitory in nature and down-regulates sterol regulatory element-binding protein (SREBP)1c. SERBP1c is a regulator of genes’ expression involved in de novo lipogenesis, fatty acid synthetase, stearoyl coenzyme A desaturase, and acetyl coenzyme A carboxylase [153]. In a randomized, multicenter, placebo-controlled phase II FLINT (Farnesoid X nuclear receptor ligand obeticholic acid for non-cirrhotic, NASH trial), 25 mg/day of obeticholic acid (OCA) for 72 weeks was associated with improvement in ≥2-point decrease in NAS without worsening of fibrosis in 45% of the patients compared to 23% of those in the placebo group (relative risk = 1.9, *p* = 0.0002) [154]. However, decrease in HDL was noted with obeticholic acid treatment. In addition, pruritus was noted in 23% of the treated patients compared to 6% in the placebo arm (*p* < 0.0001). The FDA currently has a black box warning regarding the use of OCA in patients with primary biliary cholangitis (PBC) and Child–Pugh class B or C or decompensated cirrhosis. When dosed incorrectly, OCA can lead to hepatic decompensation and liver failure in these patients. A randomized, placebo-control, double-blind phase II trial showed that Cilofexor, a non-steroidal FXR agonist, was associated with improvement in NASH. A greater than 30% reduction in hepatic steatosis by MRI-PDFF was observed in 39% of patients receiving 100 mg of cilofexor (*p* = 0.011 vs. placebo) compared to 14% of patients taking 30 mg (*p* = 0.87 vs. placebo) [155]. The benefit and risk balance of this class of medication needs to be carefully evaluated.

The liver plays a central role in lipid metabolism. Cardiovascular causes are the leading cause of death in patients with NASH. A few lipid-lowering agents have found application in the treatment of NASH. Statins have been widely studied and used as therapeutic agents in NASH. Despite initial concerns of drug-induced hepatotoxicity, their safety in patients with NAFLD has now been well established [156]. Saroglitazar magnesium, a first in class dual PPARα/γ agonist, was approved in India in March 2020 for treatment of NASH [157]. Several trials are underway in the United States. The EVIDENCES II, a phase 3, multi-center, placebo-controlled trial of 102 patients reported that saroglitazar 4 mg was associated with significant improvement in hepatocyte ballooning, steatosis, and lobular inflammation [158]. Currently, saroglitazar is being further evaluated in a clinical trial with three different dosing regimens at multiple centers in the United States. Full results are highly awaited [159].

Ezetimibe is an inhibitor of intestinal cholesterol absorption. Its use was evaluated in the MOZART trial, which compared ezetimibe 10 mg given for 24 weeks to placebo in 50 patients with biopsy-proven NASH. Although an improvement in liver fat measured by magnetic resonance imaging-derived proton density fat fraction (MRI-PDFF) was seen with ezetimibe, it was not superior to placebo and did not affect histological features [160].

The discovery of the implication of cytokines such as transforming growth factor-beta 1 (TGF-β1), platelet-derived growth factor, and angiotensin II in the pathogenesis of NAFLD led to a study evaluating the use of angiotensin II receptor blockers for exploration of a potential treatment option. Losartan, at a dose 50 mg/day, significantly reduced blood markers of hepatic fibrosis and ALT levels. Repeat biopsies at 48 weeks of therapy showed an improvement in inflammation in five patients and a reduction in fibrosis in four patients [161]. Further large-scale trials are required to substantiate these beneficial effects.

### 6.3. Effects of NAFLD on Response to Antiviral Treatment for Chronic Hepatitis B

There have been conflicting results regarding the response to antiviral treatment among patients with chronic hepatitis B and NAFLD. Some studies reported no statistical difference in virologic response in patients with or without NAFLD treated with either pegylated interferon alpha-2a or nucleotide analogues [162,163]. On the other hand, in a retrospective study on 334 CHB patients treated with entecavir or tenofovir, Kim et al. concluded that HBeAg loss was higher in patients without hepatic steatosis as measured by CAP (*p* = 0.022) [164]. The difference in virological response was thought to be due to a reduction in the contact area between drug and hepatocytes, leading to a decrease in drug bioavailability. Further detailed investigation and prospective studies are needed to address these controversial findings.

## 7. Summary and Future Research Directions

Chronic hepatitis B and NAFLD are both common causes of chronic liver disease and their coexistence has become an endemic health problem. The pathogenesis of concomitant HBV and NAFLD is complex and requires dedicated research efforts to understand the impact of each entity to the overall liver injury and carcinogenesis. While there are a large number of novel serum biomarkers and radiological methods are available or in the development for NAFLD, many are not accessible in resource-limited clinical settings and have not been carefully validated in ethnically divergent patient populations. While it is critically important to have noninvasive tests to identify the presence of NAFLD among patients with hepatitis B, the accuracy and reliability of NAFLD diagnostic modalities needs to be independently evaluated in patients with another liver condition such as chronic hepatitis B. Similarly, there are a number of therapeutic challenges in patients with coexisting NAFLD and HBV infection. The AALSD and EASL guidelines provide specific treatment recommendations for hepatitis B on the basis of ALT levels, among others. ALT elevation can be contributed from both NAFLD and HBV infection; however, no specific recommendations for HBV treatment in the setting of concurrent NAFLD and hepatitis B. There are some controversial observations that NAFLD may influence the virological response of HBV antiviral therapy. This could have significant clinical implications and deserves careful confirmation. Equally important is the efficacy and side effects of the therapeutic agents for NAFLD that need to be studied and validated among patients with both hepatitis B and NAFLD. As stated, there are many unmet medical needs with this new endemic of concomitant HBV and NAFLD; there are also new technologies and novel approaches to equip us to solve these challenges and to provide timely effective management to these patients.

## Figures and Tables

**Figure 1 microorganisms-08-01526-f001:**
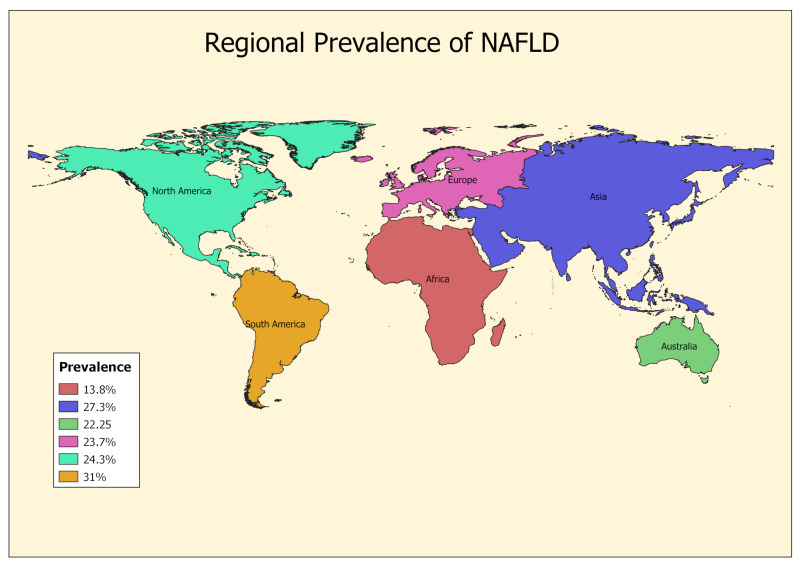
Regional prevalence of non-alcoholic fatty liver disease (NAFLD) diagnosed by radiological imaging (Pubmed and MEDLINE database from 1989 to 2015). Pooled prevalence was calculated by weighting the individual study estimates by inverse of their variation [11,20] (top panel). Global prevalence of chronic hepatitis B. Regional hepatitis B envelope antigen (HBeAg) prevalence estimates were produced from country-specific estimates. 95% CI were obtained from country-specific variances of the prevalence estimates from logistic regression model (bottom panel). Copyright permission taken from [21].

**Table 1 microorganisms-08-01526-t001:** Serum biomarkers for hepatic steatosis.

Serum Biomarkers	Description	Accuracy	Feasibility	Limitations
Fatty Liver Index (FLI)	BMI, WC, TG, GGT	Moderate AUROC: 0.84	High	Suboptimal reference standard—ultrasound is operator-dependent, insensitive to mild steatosis
Hepatic Steatosis Index	AST: ALT ratio, BMI, female sex, and DM	Moderate AUROC: 0.81	High (common parameters included)	Same as above
NAFLD Liver Fat Score	MetS, Type 2 Dm, fasting insulin, ALT and AST/ALT ratio	Very good AUROC: 0.86–0.87	Intermediate	Inclusion of serum insulin—not a routine test
Steato Test	Six components of fibrotest, Acti Test plus BMI, cholesterol, TG, glucose adjusted for age and sex	Moderate AUROC: 0.79–0.80	Intermediate	High cost, not available in all regions and clinical settings
NAFLD Ridge Score	ALT, HDL-C, TG, HbA1C, WBC, HTN	Good AUROC: 0.87	Limited to research setting	Low PPV (69%)

**Table 2 microorganisms-08-01526-t002:** Serum biomarkers for nonalcoholic steatohepatitis (NASH).

Serum Biomarkers	Description	Accuracy/Advantages	Limitations
Apoptosis markers	CK18 fragments, total cytokeratin, sFAS	- CK18 most validated, available commercially, correlates with histological improvement	- Less accurate alone - Optimal cut-offs uncertain - Poor sensitivity
Inflammatory markers	CRP, TNF, IL8, CXCL10	- Poor - Commercial assays available - Correlate with inflammatory activity in NASH	- Influenced by systemic inflammation, not validated as diagnostic markers
Lipid oxidation products	11-HETE, 9-HODE, 13-HODE, 13-oxo-ODE, LA-13-HODE, 11,12-diHETrE	- Good to excellent accuracy in small studies	- Need further validation - Require mass spectroscopy
Adipocytokines and hormones	Adiponectin, leptin, resistin, visfatin, RBP4, FBP4, FGF21	- Majority are commercialized assays - FGF21 dynamic to changes in NAFLD over time	- Mostly validated in bariatric population - Limited accuracy in isolation
Lysosomal enzymes	Cathepsin D	- Can track changes in NAFLD over time - Commercial assay available	- Limited validation - Varied interpretation in children and adults
Combined panels	NASH test, NASH diagnostic panel	- Moderate to high accuracy - Reliable - Available commercially	- High cost - Validation only in bariatric population

**Table 3 microorganisms-08-01526-t003:** Noninvasive biomarkers for hepatic fibrosis.

Serum Biomarkers	Description	Accuracy	Limitations
AST/ALT Ratio	AST and ALT	AUROC: 0.66–0.74 for F3 fibrosis Sensitivity (Sn): 40% Specificity (Sp): 80%	Low sensitivity Non reproducible as ALT may change due to the presence of NAFLD or HBV infection
AST/Platelet Ratio Index	AST and platelet count	AUROC 0.74 for F3 fibrosis Sn: 65%, Sp:72%	Low accuracy
Fibrosis-4 Index	Age, AST, ALT, and platelet count	AUROC0.80 for F3 fi fibrosis Sn:65%, Sp: 97%; by dual cut-offs	Low accuracy
NAFLD Fibrosis Score	Age, BMI, impaired fasting glucose, and/or DM, AST, ALT, platelet count, and albumin	AUROC 0.75–0.82 for F3 fibrosis Sn: 73%-82%, Sp: 96-98% by dual cut-offs	Interpretation of BMI might differ across different ethnic groups Use of ALT may confer variable results
BARD Score	AST, ALT, BMI, and diabetes	AUROC 0.69–0.81 for F3 fibrosis Sn: 62%, Sp: 66%	Interpretation of BMI might differ across different ethnic groups

**Table 4 microorganisms-08-01526-t004:** Comparison of magnetic resonance imaging-proton density fat fraction (MRI-PDFF) and controlled attenuation parameter (CAP) for diagnosis of steatosis [99].

Grades of Steatosis		AUROC (CI)	*p*-Values
Grade 1–3 versus Grade 0	MRI-PDFF	0.99 (0.98–1.00) 0.85 (0.75–0.96)	< 0.01
	CAP		
Grade 2–3 versus Grade 0–1	MRI-PDFF	0.90 (0.82–0.97) 0.70 (0.58–0.82)	< 0.01
	CAP		
Grade 3 versus Grade 0–2	MRI-PDFF	0.92 (0.84–0.99) 0.73 (0.58–0.89)	0.02
	CAP		

**Table 5 microorganisms-08-01526-t005:** Comparison of magnetic resonance elastography (MRE) and transient elastography (TE) for diagnosis of fibrosis [99].

Stages of Fibrosis		AUROC (CI)	*p*-Values
Stage 1–4 versus Stage 0	MRE	0.82 (0.74–0.91) 0.67 (0.56–0.68)	0.01
	*Fibroscan		
Stage 2–4 versus Stage 0–1	MRE	0.89 (0.83–0.86) 0.86 (0.77–0.95)	0.46
	Fibroscan		
Stage 3–4 versus Stage 0–2	MRE	0.87 (0.78–0.96) 0.80 (0.67–0.93)	0.19
	Fibroscan		
Stage 4 versus Stage 0–3	MRE	0.87 (0.71–1.00) 0.65 (0.45–0.94)	0.05
	Fibroscan		

***** Transient elastography (TE).

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
