# Peer review of "A New Endemic of Concomitant Nonalcoholic Fatty Liver Disease and Chronic Hepatitis B"

_microorganisms, 2020, doi:10.3390/microorganisms8101526_

Round 1
Reviewer 1 Report
September 20, 2020
Review for Microorganisms, the manuscript “A new Endemic of Concomitant Nonalcoholic Fatty Liver Disease and Chronic Hepatitis B”
General Comment
This is an excellent review on the gradually emerging entity. Indeed, it has been noticed over the recent years that patients with chronic hepatitis B (mostly on antiviral therapy) are noted to have an increasing incidence of fatty liver during the long-term follow up.
Now the authors have presented this new endemic of the combination of two diseases. Authors are commended for their exceptionally comprehensive review on every possible aspect of this entity (combination of CHB with NAFLD) including possible pathogenesis and future management. Many physicians and the healthcare workers will benefit from this outstanding review and will be able to provide the proper treatment for those with concomitant CHB and NAFLD.
Author Response
We like to thank the reviewer for the encouraging and enthusiastic comments.
Reviewer 2 Report
In recent years, non-alcoholic fatty liver disease (NALFLD) have replaced chronic hepatitis C virus infection as the most common chronic liver disease in the western world. NAFLDs cover a wide spectrum of liver diseases, ranging from non-alcoholic-fatty liver (NAFL), non-alcoholic fatty liver hepatitis (NASH) and secondary fatty liver to fatty cirrhosis. Various factors are associated with a higher risk of developing NAFLD to fatty cirrhosis. These include insulin resistance, central obesity, genetic and environmental factors and changes in the intestinal flora. The surge in NAFLD is closely linked to pandemic increase in metabolic multisystem diseases. NASH has a multifactorial etiology, in which genetic and lifestyle facrors ( over-and under nutrition) contribute to excessive fat accumulation, mitochondrial dysfunction, endotoxins, pro-inflammatory cytokines to chronic inflammation. NASH is per se considered a risk factor for the development of cirrhosis and hepatocellular carcinoma (HCC). Chronic hepatitis B and C are an important morbidity and mortality factor. They are one of the leading causes of death worldwide with more than 1,5 million deaths. Against this backround, studies of clarify the simultaneous existence of NAFLD and hepatitis B or C are great epidemiological, medical and social relevance and topically. The pathogenetical mechanisms of liver damage underlying the diseases, such as progressive inflammation and fibrosis, synergistically lead to further progressive hepatocellular damge up to cirrhosis and HCC. Comprehensive clinical, laboraty, imaging and histolgical diagnosis is required to clarify ensure the joint existence of these two diseases. Liver biopsy is still the gold standard for diagnosis of NAFLD or NASH. However, as an invasive method, it is associated with complications in very small percentage of cases. After a short introduction, the epidemiological situation of NAFLD and hepatitis B and the presentation of the coexistence clinical manifestations of both diseases, the authors give an overview of the current availability of serum markers for the identification of NAFLD in known chronic hepatitis B. These biomarkers will be used to detect fatty liver, NASH and fibrosis assessment of severity. They conclude that are several markers and procedures available, but that they are ultimately immature, too expensive or not sufficient meaningful. Most biomarkers for fatty liver, inflammation and fibrosis in NASH have not been validated for patients with concurrent chronic hepatitis B. The same ultimately applies to detection of fatty liver. The authors note that all these methods must first be assessed on large cohort of patients by comparing liver biopsy and histological examination. The authors see great promise in the investigation of gene polymorphisms and miRNAs as possiblle biomarkers. In subsequent section, the possibilities of abdominal ultrasound and MRI respectively abdominal and MRI elastography are presented. The ultrasound elastography is recommened as a method of fibrosis assessment in NAFLD. In further section the therapeutic options for patients with NAFLD and chronic hepatitis B are presented. These include lifestyle changes, in individual cases, bariatric therapy and the wide range of different diabetes therapy agents such as pioglitazone, sitagliptin, metformin and lipid therapy such as statins and ezetimibe. Of particular interest is obetichol acid, a farnesoid receptor agonist, which in initial studies showed an improvement in NASH. Because of a severe pruritus, also known from PBC patients, and danger of hepatic decompensation when the dosage is too high, a decision on this drug is ultimately stil pending. Finally, the effects NAFLD on the antiviral therapy of chronic hepatitis B are pointed out.
The paper gives a good overview of the current state of knowledge of the joint existence of NAFLD and hepatitis B. The statements are factual and are well documented by a very extensive literature ( 166 citations). The good content is contrasted by the confusing layout of the individual sections, which makes if difficult th read the work. This should be changed in the final version. The bibliography must be corrected as a whole and designed according to the journal`s guildelines. Individual minor errors in the should also be corrected.
The work is written in correct English. The illustrations and tables complement the text well. I commend that you accepte the work after correction.
Author Response
We thanks reviewer for the careful appraisal of our manuscript.
We revised the headings and subheadings to clearly separate the individual sections. Hopefully, it is more cohesive and easier to read.
We will leave the formatting of the references to the editorial staff for now but will revise if necessary.